# God as Highest Truth According to Aquinas

Enrique Martínez 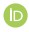

Department of Education and Humanities, Universitat Abat Oliba CEU, 08022 Barcelona, Spain;
emartinez@uao.es

**Abstract:** Contemporary public opinion has come to assume that we live in the post-truth era, in which judgments on the most relevant realities of human life have been left in the hands of mere emotions. In such a context, it is very opportune to redirect our gaze toward the concept of truth, in order to help to adequately ground such a primordial reality as that of the personal being. Furthermore, this is the object of the present research, following the thought of St. Thomas Aquinas. To this end, we attempt to argue that the primacy in the analogical significance of the truth corresponds precisely to the person, as a subsistent being whose esse is intelligible to himself. Following the analogical ascent, we consequently arrive at God, who is absolutely intelligible to himself. We have to conclude, therefore, that the personal God is the highest truth. As a corollary to this argument, we add that the perfective dynamism of the personal life is realized in an eminent way in the communication of truth through words, also in God.

**Keywords:** person; truth; God; analogy; intelligible; universal; knowledge; metaphysics; Thomas Aquinas; Thomistic School of Barcelona

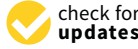



## 1. Introduction: Wondering about God in a Post-Truth Era

Contemporary public opinion has come to assume that we live in a post-truth era (Ibáñez 2017). A consequence of this is that judgment about the most important realities of human life are left in the hands of mere emotions (Belmonte 2020). Is this post-truth age affecting how we understand the personal being? From the fruitful encounter between faith and reason, Western culture has articulated its worldview of human life based on the concept of person (Martínez 2010). However, nowadays, we talk about "non-human persons" when referring to orangutans, cyborgs, etc.

In such a context, it is opportune to redirect our gaze toward the concept of truth, in order to adequately ground such a primordial reality as that of the personal being. Furthermore, this is the object of the present research.

To this end, we attempt to argue that the primacy in the analogical significance of the truth corresponds precisely to the person, as a subsistent being whose esse is intelligible to himself. Following the analogical ascent, we consequently arrive at God, who is absolutely intelligible to himself. We have to conclude, therefore, that the personal God is the highest truth. As a corollary to this argument, we add that the perfective dynamism of the personal life is realized in an eminent way within the communication of truth through words, also in God.

Following Saint Thomas Aquinas, this thesis has been defended in the Thomistic School of Barcelona mainly by Jaime Bofill and Francisco Canals (Forment 1988, 1998). This position follows the line of interpretation of the Thomism of John Capreolus, Louis Billot, Étienne Gilson or Cornelio Fabro, which identifies the esse as the formal constituent of the person. Different positions are those of Thomas Cajetan or Jacques Maritain, more essentialist, or those of the transcendental Thomism of Johannes Baptist Lotz, which identifies the formal constitutive in consciousness (Contat 2013). In addition, we must refer here to Saint Bonaventure in order to acknowledge another way of ascending to God (Lázaro 2019).

Summarizing what we wish to develop later regarding this issue, Canals (1987, p. 576) states the following:

> Not only in the line of the transcendental good (...) but also in the line of the transcendental truth, we must affirm the primacy of the spiritual subsistent, of the person. Only the person is capable not merely of the language about what things are, but also of being the one to whom one can say what the spirit is originally and constitutively capable of saying (...) We only talk to persons.

## 2. The Primacy of the Universal in the Intelligible Order

The thesis set out by Canals consists, therefore, of affirming that the person, in his singularity, comes first in the order of transcendental truth. However, we must begin with a considerable objection: it seems that what is true must be said first of the universal and not of the person, which is something singular. This is what we read from Saint Thomas Aquinas (1889, I, q.12, a.8 ad 4) himself:

> The natural desire of the rational creature is to know everything that belongs to the perfection of the intellect, namely, the species and the genera of things and their types, and this everyone who sees the Divine essence will see in God. But to know other singulars, their thoughts and their deeds does not belong to the perfection of the created intellect, nor does its natural desire tend to these things.

Jaime Bofill (1950, p. 231) laments, "an unfortunate interpretation of this passage has led many authors to conceive the intellectual goal in the horizontal level of generalizing abstraction."

Before solving the objection, it seems opportune to know how the primacy of the universal has been established in the order of the intelligible.

Moreover, we consider that the affirmation of the aforesaid primacy is the result of judging as belonging to knowledge, as such that which is only specific to Man's knowledge. Let us see why.

The corporeal condition of the human person necessarily involves limitations that leave his intellect in potency with respect to the intelligible. How then does human intellect reach the actuality of knowing?

We can identify various solutions. Thus, for some, the intelligible is found in the same singular corporeal being, which is then intelligible in itself prior to the act of knowing—now, this is the case in Scotistic and Nominalist entitative intuitionism. By contrast, for others, it is necessary to take the intelligible to the intellect, separating it from the singular corporeal being, with a universal species resulting from this separation; this one would be known either intuitively –from the standpoint of rationalistic and idealistic eidetic intuitionism—or through representation—within Aristotelian–Thomistic conceptualism.

However, it seems as if certain universal agreement has been generalized inasmuch as that intuition is the way that corresponds to the essence of knowing as such, together with a rejection of the concept as representation, thus accusing it of substituting and moving away from reality. Henri Bergson (1946, pp. 145–46), for example, says:

> Here is a point upon which everyone will agree. If senses and consciousness had an unlimited scope, if in the double direction of matter and spirit the power of perceiving was unlimited, there would be no need to conceive nor to reason.

This way the object could be reached in the intelligibility that it would have in itself prior to the act of knowing. We read this, for instance, in Francisco Suárez (1856, III, c.5, n.6):

> And if it is argued that such a specie is required as a "substitute" instead of the "object", this will easily be refuted, because the object, and therefore everything that substitutes it, precedes the act of knowing: then it cannot be produced by it.

This primacy of intuition, which would be the proper way of knowing as such, leads to affirm—except in Nominalism—the primacy of the universal in the order of the intelligible, which, not to depart from reality, is intuitively known. In other words, we once again

encounter the Platonic world of ideas, where "idea" means "what is seen". Canals (1987, p. 136) offers us a succinct relation of these forms of eidetic intuitionism that consecrate the primacy of the universal:

> Hence, the immediatist postulate, which we have seen before functioning in the disintegration of all essence, has also been exerted as a requirement to attribute the character of true reality, and only truly known, that is, intuited, to the contents of the intellect, to the essences and the *ideas* –that is, the realities *seen*–, while recognizing as the most proper *seeing* or *perceiving* the capturing of them. From the *being* contemplated by the Eleatics, and even from the numbers and figures of the Pythagoreans; from the intelligible world constituted by the hierarchy of ideas, *genera* and *species* of the Platonists; even the *clear and distinct ideas* of Cartesianism, and even the *essences* or the *values* grasped through immediate intuition, according to modern Phenomenology; the intuitionist postulate has worked in all these cases to affirm the immediate patency of cognitive contents, universally attainable by every man who has truly achieved knowing.

This brief state of question presents a primacy of the universal in the intelligible order, which goes hand in hand with the primacy of intuition. This way, what is proper to Man's way of knowing due to his corporeal limitation—that is, generalizing abstraction—has ended up being postulated as corresponding to knowing as such. In conclusion, universality has become the formal reason of intelligibility.

### 3. The Essence of Truth and Reason of Intelligibility

Next, the approach set out above is contrasted with St. Thomas Aquinas' metaphysics of knowledge, in an attempt to determine whether, according to the latter, universality is the formal reason of intelligibility.

First, it is worth recognizing that the aforesaid primacy of intuition, and the universal is explained ultimately by having forgotten that the esse or act of being is act and perfection, as Domingo Báñez (1585, I, q.4, a.1 ad 3) here states:

> And this is what Saint Thomas most frequently claims and what Thomists do not want to hear: that the esse is the actuality of every form or nature and is not found in anything as container and perfectible, but as received and perfective of that in which it is received.

Indeed, that the esse is act and perfection allows us to understand knowing, formally, as a certain esse. This way, whatever belongs to the essence of knowledge as such, including the reason of intelligibility, should also be understood as act and perfection. In contrast, forgetting this fundamental metaphysical principle is what leads us to judge knowledge and intelligibility only from the perspective of potentiality.

We can see this in Saint Thomas Aquinas, identifying in his work the different meanings of "truth", that is, of intelligibility.

The first of these meanings is, precisely, the esse as an act, as the foundation of all truth or intelligibility. Saint Thomas Aquinas (1976, q.1, a.1 in c.) states:

> [Truth is defined] according to that which precedes truth and is the basis of truth.
> This is why Augustine writes: *The true is which is*.

A second meaning refers to every being that, because it has the form or nature that corresponds to it, it is "true" or intelligible, namely, adequate to be understood (Aquinas 1889, I, q.16, a.2 in c.): "Everything is true according as it has the form proper to its nature". According to this second meaning, it is also said that the intellect—as a knowing being—is true or intelligible inasmuch that it has the form that corresponds to it, which is the species or likeness of the known thing:

> It is necessary [continues the aforementioned text] that the intellect in so far as it is knowing, must be true, so far as it has the likeness of the thing known, this being its form (...)

Finally, according to a third meaning, we refer to "true" as the conformity between the known form and the thing to which it is similar:

> ( . . . ) as knowing [continues the same text]. For this reason, truth is defined by the conformity of intellect and thing and hence to know this conformity is to know truth ( . . . ) yet it does not apprehend it by knowing of a thing *what a thing is*. When, however, it judges that a thing corresponds to the form, which it apprehends about that thing, then first it knows and expresses truth.

This third meaning is what is signified by the definitions given by Aquinas (1976, q.1, a.1 in c.), quoting Saint Hilary and Saint Augustine; that is, truth as a manifestation of the esse:

> In a third way, what is true is defined as the effect that follows, and Hilary defines it thus, saying that *what is true is what manifests and declares the esse* and Augustine, in the book *De Vera Religione*, says that *the truth is that through which what is, is revealed*.

Therefore, the most proper meaning of "truth" is given in saying what a thing is and not in the apprehension of the species.

Assuming these distinctions of the different meanings of "truth" in Aquinas, we can now ask ourselves about the reason of the intelligibility of the being—adequate to be understood—as well as that of the species in the intellect—adequate to be known in its conformity with the thing. Furthermore, the answer lies in the first meaning of "truth": the esse in act. Saint Thomas Aquinas (1889, I, q.16, a.1 ad 3) says:

> In the same way, the *esse* of the thing, not its truth, is the cause of truth in the intellect. Hence, the Philosopher says that a thought or a word is true *from the fact that a thing is, not because a thing is true*.

Furthermore, there is no universal subsistent, and we must deny that the formal reason of the intelligibility of the being is universality, not the actuality of the esse. For this reason, the unintelligible corresponds to the potential; hence, as the most potential thing is matter, that is why human intellect must abstract from the material to be able to know, resulting then the universal in prædicando. Then, the reason of intelligibility is not in the universality but in the immateriality proper to the actual:

> The universal [Canals (1987, p. 573) states], precisely because it is what can be said regarding many subjects, and for that reason something that in itself, in its direct intelligibility, does not properly exist nor is apt to exist, is not intelligible because of its universality, but only because of its immateriality.

Ergo, it should not be said that something is unintelligible due to its singularity, but as it has been stated, due to its materiality. Notice this important thesis regarding the reason of the intelligibility of the being in the following two remarkable texts from Aquinas (1996, q.un., a.2 ad 5; 1889, I, q.56, a.1 ad 2):

> *The first text:* The human soul is a certain individuated form; and equally its potency, which we call possible intellect, and the forms understood in act; for something is understood in act because it is immaterial, and not because it is universal; but rather the universal has to be intelligible because it is abstracted from the individuating material principles. However, it is evident that separate substances are intelligible in act, and yet are certain individuals.

> *And the second text:* Of the singulars that exist in bodily things, there is no intellection in us, not because of singularity, but because of matter, which is the principle of their individuation. Therefore, if there are some singular subsistents without matter, such as angels, nothing prevents them from being intelligible in act.

## 4. The Analogical Ascent to God, Highest Truth

In consonance with the previous two texts, "the separated substances are intelligible in act, and yet are certain individuals", and "as far as singular subsistents without matter

is concerned, such as angels, nothing prevents them from being intelligible in act". These statements lead us to the central point of this thesis.

According to the conclusion we have reached regarding the reason of intelligibility, we can state that angels are intelligible in act because of the actuality of their esse, excluding all potential materiality. Thus, it is because of this immateriality that they can return to their own essence and know themselves in the actuality of their singular esse (Echavarría 2013). Aquinas (2000, q.un., a.1 ad 12) says, with remarkable audacity:

> If an ark could be subsistent by itself without matter, it would understand itself, because the immunity from matter is the essential reason of intellectuality.

In a sense, this singular can be said to be universal, but in causando, not in prædicando, inasmuch as a being is more universal in its causality as it is more of an act (Aquinas 1971, pr.): "The separated substances are universal and the first causes of esse." That is why God, Pure Act, is the universal uncaused Cause.

However, this intelligibility must also be said of human intellect, given its immateriality, although it is less than that of angels and much less than that of God. Therefore, St. Aquinas (1889, I, q.86, a.1 ad 3) states, "if there be an immaterial singular such as the intellect, there is no reason why it should not be intelligible." Of course, the corporeal condition of the human person implies that his intellect is in potency to know—and to know itself—having to pass to the act of understanding through the abstraction carried out by the active intellect, which is nothing but the same intelligible light of the subsistent soul itself. Canals (1987, p. 575) explains this in the following manner:

> If intelligibility is constituted by immateriality and not by universality, that is, the universal is intelligible because it is immaterial, it is understood that St. Thomas can attribute the character of intelligibility to the spiritual and existing singular, despite the contingency, facticity and temporality, which characterizes all spiritual realities, as it is immediately experienced by human consciousness.

Therefore, if we propose a proportional gradation of intelligible realities, we must say that the most intelligible is the most actual and immaterial, namely the singular intellectual subsistent. Below this, and because of the dependence of matter, the intelligible is in a situation of potentiality; it only becomes intelligible in act in the same knowing of the intelligent precisely thanks to the actuality of its esse. In fact, there is no truth if there is no intellect. Referring this to God, Aquinas (1889, I, q.16, a.7 in c.) states: "If there were no eternal intellect, there would be no eternal truth either".

Now, every intellectual subsistent that knows and loves itself in its own esse and makes the intelligible in act has received since ancient times a name of singular dignity, which is that of "person". Aquinas (1889, I, q.29, a.1 in c.) says:

> Particular and individual find themselves in a much more specific and perfect way in rational substances, which dominate their acts, being not only moved, like others, but also acting by themselves. Actions are in singulars. Thus, among all substances, singular substances of a rational nature have a special name. This name is *person*.

Thus, it must be recognized that the person is a good beloved in himself with a love of friendship. Indeed, humankind is not loved; rather, Peter or John are. Therefore, the personal being occupies the primacy in the analogical scale of the good.

In addition, the person is intelligible to himself by the subsistent actuality of his esse, on which it depends that the intelligible in potency is made intelligible in act. Then, the intellectual subsistent also possesses primacy in the analogical scale of what is true:

> This thesis [concludes Canals (1987, p. 576)] leads us to conceive the truth, the immaterial perfectivity of the being, as an *esse* in act, a perfection and ontological plenitude, of which immaterial subsistents or personal beings fully participate. Personal beings possess the truth in a different and more perfect way than that in which beings that are outside the order of the immaterial and intellectual, of the

intelligible order, possess it, and are only intelligible in potency by extrinsic and objective intelligibility.

However, God is called "person" in the highest degree (Aquinas 1889, q.29, a.3 in c.):

Person means what is most perfect in all nature, that is, the subsistent in rational nature. Therefore, since everything than belong to perfection must be attributed to God, because his essence contains in itself all perfection, it is fitting that God should be given the name person. However, not in the same sense in which it is given to creatures, but in a more sublime way.

Furthermore, this must be affirmed in the natural light of reason by means of analogical ascent (Roszak 2017). Even assuming that God reveals himself; therefore, revelation is a rational operation. For this reason, Saint Thomas Aquinas (1889, I, q.2, a.3 s.c.) refers to the person of God—*ex persona Dei*—when he deals with the revelation of the divine name at Sinai: "Against this is what is said in Exodus 3:14 of the person of God: *I am who I am*" (Roszak 2015).

To sum up, the intelligibility for itself, proper to the intellectual subsistent, must be said eminently to refer to the personal God, who is therefore the supreme intelligible truth to himself.

## 5. The Communication of Personal Truth

Let us conclude with a corollary. We must recognize in the actuality of the esse an inclination to communicate its own perfection (Aquinas 1965, q.2, a.1 in c.): "The nature of any act consists of communicating itself as much as possible. From which it derives that each agent acts insofar as it is in act". Therefore, it is inferred that the personal being, aware of itself by the actuality of its subsistent esse, tends to be communicative and manifestative of what it is; and the means for doing so is no other than the true word that says what is known (Martínez 2013).

Moreover, only the personal being is likewise the recipient of that true word, the one to whom another can speak:

Only the person is capable [Canals (1987, p. 576) states] not merely of language about what it is, but also of being the one *to whom* one can say what the spirit is originally and constitutively capable of saying. All language with intelligible meaning, all vital communication at the level of personal spirit, in the line of knowledge of reality as such, and in the line of moral action, or interpersonal social coexistence, as well as in the line of rational efficiency or technical performance is, of itself, exclusively destined to be received, consciously and intelligibly, by personal beings. We only talk to persons, in the theoretical or moral order, in the normative order of political life, in the regulative or evaluative order of a rational efficiency.

This way, and in the face of the disintegrating post-truth of human life, we need to state that the perfective dynamism of the human person occurs in the communication of truth through word. Moreover, it is thanks to this that friendship is possible; indeed, the affective union between friends is only understood in the light of a real union, rooted in the communicative word of one's own personal life (Cortés 2016; Martínez 2012). Referring to truth as the foundations of such a recognition of the dignity of the personal being, Jaime Bofill (1950, p. 165) states:

Therefore, loneliness is definitively overcome, as our aspirations to be understood, appreciated, loved, as well as those of an opposite direction, are equally satisfied, to pour into others the fullness of our heart in peaceful confidence. Through them, Man is situated in his true environment, namely the family and society, and occupies his place in the Universe. The measure of this perfection and of the corresponding joy will become visible for us by the consideration of what it means: the enrichment of a Person through what is most valuable in the entire Universe, namely another Person, to surrender oneself not in some of its aspects

or goods, more or less external, but by introducing us into the intimacy of his life and being.

Beforehand, it has been stated that divine revelation is an operation proper to a personal Being (Roszak 2016). Therefore, since the fullness of revelation is given in the Incarnation of the divine Word, Saint Thomas Aquinas (1972, c.15, lect.3; Martínez 2015) explains that friendship with Christ consists in entering into communion through the communication of Truth, which is his own Person:

> Here he places the true sign of friendship on his part, which is *what I heard from my Father, I let you know*. Indeed, the true sign of friendship is that a friend reveals the secrets of the heart to his friend. Indeed, because the hearts of friends are one single heart and one single soul, it does not seem that a friend places outside of his heart what he reveals to his friend (...) For God, making us sharers of his wisdom, reveals to us his secrets: *communicating to the holy souls through the nations constituted friends of God and prophets*.

Accordingly, we can quote here what Pope Benedict XVI (2009, p. 642) concisely expressed at the beginning of his encyclical Caritas in veritate: "In Christ, *charity in truth* becomes the Face of his Person, a vocation to love our brothers in the truth of his plan. Indeed, he himself is the Truth".

**Funding:** This research received no external funding.

**Institutional Review Board Statement:** Not applicable.

**Informed Consent Statement:** Not applicable.

**Data Availability Statement:** Not applicable.

**Acknowledgments:** Special thanks to the people who help me in the preparation of this paper: Alessandro Mini, Arturo González de León, Filomena de la María, and Vanesa Berlanga.

**Conflicts of Interest:** The author declares no conflict of interest.

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
