# Peer review of "God as Highest Truth According to Aquinas"

_religions, doi:10.3390/rel12060429_

Round 1

Reviewer 1 Report

While this is a clear bit of Thomistic exposition, there's nothing original here. The author is at least up front about their framework: "We will have to conclude, therefore, that the personal God is the highest truth. As a corollary to this argument, we will add that the perfective dynamism of the personal life is realized in an eminent way in the communication of truth through words, also in God."

That's all well and good for those who are committed Christians, who want to take as their starting point that God not only exists, but that he has being exactly as Aquinas described. It also only works if one has a highly reductive view of what "truth" is, rather than an actual familiarity with post-modern theories instead of the caricature of what these theories represent as indicated by the quote from former pope Benedict XVI who stated: "building a dictatorship of relativism that does not recognize  anything as definitive and whose ultimate goal consists solely of one’s own ego and desires”. This isn't what post-modern theories do.

In my view, an academic journal is a home for academic discussion, not a place for positing an unassailable faith-based worldview. Furthermore, the title of this journal is "religions" in the plural, not "Catholic viewpoints" or something. 

I would suggest that a Catholic journal or magazine would be a better home for this article. 

Author Response

The metaphysical assertion of the primacy of the person in the analogical significance of the truth is an original contribution. This thesis applies eminently to God, as a personal being. This approach is metaphysical, according to a rational argument, and it is not based on faith. The concept of truth used is the one that corresponds to a philosophical tradition extolled by Hegel. Heidegger's own critique of this concept of truth in Western metaphysics is based precisely on the primacy of being, as defended in the article. The purpose of this article is precisely the philosophical discussion with those who adopt opposite metaphysical positions in the field of Theodicy.

Reviewer 2 Report

The article makes a very important argument regarding the relationship between truth and personhood in the thought of Thomas Aquinas. As it stands the text needs substantial revision/expansion as well as editing for grammar and style.

As this referee understands it, the paper builds on the notion of truth articulated in lines 158-166. This is a crucial distinction within Thomas's thought that gets to the deepest notion of truth: the truth of being in aletheia (manifestation/revelation); self-disclosure of being in act. This is the truth we know as the truth of a person. 

The climax of the paper is the claim from Thomas quoted at lines 237-46. This is, again, a very important point given the context the author intends to address, i.e., post-truth culture. God is both the fullness of personhood and the fullness of truth. Therefore God is not an abstract truth, but a truth that is self-disclosive through its Word and mediates its being to others through friendship. The friend reveals his heart to his friends.

All of this is every good, but perhaps too compactly stated as written. This is probably an argument for an essay closer to 7000 words than the current length. This would allow the author to engage a broader segment of scholarship on the question. As it stands the sources are relatively restricted to a particular segment of scholarship on Thomas Aquinas. The author clearly indicates that he or she is engaging the Barcelona school of interpretation, but there could be greater attention to the larger conversation in Thomistic studies to support the credibility of the argument.

The writing is difficult to follow at times. A further edit for grammar and style is a must. There are clear translation issues as well. For example, 'convenient,' used here in the Thomistic sense, should really be translated as "opportune" or simply 'fitting' in contemporary English. Another important example is the use of the English "conscience"(line 231), which seems, based on the context, to be more likely translating a word that would refer to "consciousness." There are a few instances when the author's voice is not clearly distinguished from quoted material (line 181). This could be cleaned up by a further edit. The punctuation, especially the use of emdash or hyphen needs a careful edit.

Author Response

It is explained that the position of the Thomist school in Barcelona follows the line of interpretation of the Thomism of John Capreolus, Louis Billot, Étienne Gilson or Cornelio Fabro, who identifies the esse as the formal constituent of the person (Contat 2013). Different positions are those of Thomas Cajetan or Jacques Maritain, more essentialist, or those of the transcendental Thomism of Johannes Baptist Lotz, which identifies the formal constitutive in consciousness. The reference to another path other than Thomism is also added, such as that of Saint Bonaventure (Lázaro 2019).

A complete revision of the English text has been made, incorporating the reviewer's suggestions.

Reviewer 3 Report

The work presented addresses one of the most worrying issues in today's society: hyper-emotionality. The consideration of emotions nowadays has acquired an absolute value, especially since attempts have been made to endow them with an intelligence (emotional intelligence) that in truth responds to a design of emotions. This has caused the substitution of truth and the faculty of understanding by the anthropological reduction in favour of understanding. The problem needs to be addressed and this is done in the article we are evaluating. 

The author uses the philosophy of Thomas Aquinas to address this reality. In principle it seems adequate, since Thomas Aquinas presents an adequate philosophy and anthropology, looking at the background of his work De veritate.

Una vez tomada esa opción, el autor desarrolla la argumentación, las fuentes y la bibliografía con maestría, construyendo un argumento sólido y bien fundamentado tanto en el pensamiento tomista en sí, y en su contexto teológico, como a la hora de presentar una alternativa al problema que se plantea y, por lo tanto, alcanzando sus objetivos.

Once this option is taken, the author develops the argumentation, sources and bibliography with mastery, building a solid and well-founded argument both in Thomistic thought itself, and in its theological context, as well as when presenting an alternative to the problem posed and, therefore, achieving its objectives.

We believe that other response options could have been taken, not only at a global level, from other theological schools, but also in the structuring itself regarding the reflection on the ascent to God, I am thinking especially of St. Bonaventure, which the article does not mention, especially because the author also speaks of the encyclical of Pope Benedict XVI Deus caritas est and this author is very present. I think that some reference would be pertient, I suggest: Manuel Lázaro, "Más allá de la quiditas: reflexiones sobre el proyecto metafísico bonaventuriano", Cauriensia 14 (2019) 49-80. As well as vol. 2 (2007) of the same journal.

In any case, this suggestion should in no case be taken as imperative, but as part of the discussion of the article itself.

Author Response

It is explained that the position of the Thomist school in Barcelona follows the line of interpretation of the Thomism of John Capreolus, Louis Billot, Étienne Gilson or Cornelio Fabro, who identifies the esse as the formal constituent of the person (Contat 2013). Different positions are those of Thomas Cajetan or Jacques Maritain, more essentialist, or those of the transcendental Thomism of Johannes Baptist Lotz, which identifies the formal constitutive in consciousness. The reference to another path other than Thomism is also added, such as that of Saint Bonaventure (Lázaro 2019).

Round 2

Reviewer 2 Report

The revisions to this text are helpful for readers.

There remain a few places that need revisions. 

In lines 153 and 158 there appear to be interjections of the author voice into quoted material. Normally this is set off by brackets [] rather than hyphens or emdashes. Something similar occurs in the quote at line 290.

There is a style issue in the sentence at lines 277-8, namely a gerund instead of an infinitive form of the verb 'refer.' Recommended revision would read: "To sum up, the intelligibility for itself, proper to the intellectual subsistent, must be said eminently to refer to the personal God, who is therefore the supreme intelligible truth for himself."

There is a grammatical flaw in the sentence at lines 288-9. Recommended revision would read: "Moreover, only the personal being is likewise the recipient of that true word, the one to whom another can speak"

Finally, the author employs the term 'Man' in a few instances, e.g. Line 301. Generally, this does not follow the standards of style in English language academic journals. But this is a question for the journal editor not a referee. My personal recommendation would be to change those references to "human person."

On a different point, I wonder whether this journal is the right venue for this article. It would seem that scholars studying St. Thomas would be more likely to find the article if it were in English language journals like The Thomist, Communio, Gregorianum, Nova et Vetera, American Catholic Philosophical Quarterly, or other similar outlets.

Author Response

I sincerely appreciate the different comments of the reviewer. I have incorporated all of your suggestions into the article.

The last comment is about the most suitable journal for the publication of the article. My opinion is that the monographic issue on Theodicy in Religions is ideal for a metaphysical study of God from the perspective of his intelligibility. But I leave the decision to the editors.

Best regards